

**Theoretical Interpretation of the Exceptional Sediment**
**Transport of Fine-grained Dispersal Systems Associated with**
**Bedform Categories**
Tian Zhao[1], Qian Yu[1], Yunwei Wang[2], and Shu Gao[3]
[1]Ministry of Education Key Laboratory of Coast and Island Development, Nanjing University, Nanjing 210023,
China.
[2]College of Harbour, Coastal and Offshore Engineering, Hohai University, Nanjing 210098, China.
[3]State Key Laboratory of Estuarine and Coastal Research, East China Normal University, Shanghai 200062, China.
*Correspondence to*: Qian Yu (qianyu.nju@gmail.com)
**Abstract.** Being a widespread source-to-sink sedimentary environment, the fine-grained dispersal system (FGDS)
features remarkably high sediment flux, interacting closely with local morphology and ecosystem. Such exceptional
transport is believed to be associated with changes in bedform geometry, which further demands theoretical
interpretation. Using van Rijn (2007a) bed roughness predictor, we set up a simple numerical model to calculate
sediment transport, classify sediment transport behaviors into dune and (mega-)ripple dominant regimes, and discuss
the causes of the sediment transport regime shift linked with bedform categories. Both regimes show internally
consistent transport behaviors, and the latter, associated with FGDSs, exhibits considerably higher sediment transport
rate than the previous. Between lies the coexistence zone, the sediment transport regime shift accompanied by
degeneration of dune roughness, which can considerably reinforce sediment transport and is further highlighted under
greater water depth. This study can be applied to modeling of sediment transport and morphodynamics.

**1 Introduction**
Shaped by fine-grained (median grain size $d$ = 15 ~ 150 μm, i.e. 6.0 ~ 2.7 φ) bed, the fine-grained dispersal system
(FGDS) is a type of sedimentary environment that is rooted in coastal, riverine, deltaic, marine, and subglacial systems,
as well as characterized by remarkably high total sediment fluxes (Ma et al., 2017). In science and engineering
disciplines, FGDSs are of great importance because they are crucial source-to-sink systems, highlighting the unique
role that suspended sediment transport processes play in developing phenomenal sediment transport. Generated from
erosion in sources (mountains and riverbeds), transported as suspended load, and eventually preserved at sinks (coastal
zone, continental shelves, and deep seas) (Kuehl et al., 2016; Leithold et al., 2016), riverine fine sediments increase
turbidity of estuarine and nearshore waters, forming mud depositional systems of considerable thickness (Gao &
Collins, 2014; Wright, 1995). The source-to-sink processes these sediments undergo not only notably alter local
sediment dynamic environments (Wright & Nittrouer, 1995), material cycling processes (Blair & Aller, 2012; Kuehl
et al., 2016), and ecosystems (Venkatesan et al., 2010), but will shape a distinctive sedimentary system over a long




time span as well (Gao & Collins, 2014). In addition, knowledge of FGDS will be of great benefit to tackling real-
time engineering issues, including navigation channel dredging (van Maren et al., 2015), harbor construction
(Winterwerp, 2005), monitoring morphological responses of tidal flat reclamation (Lee et al., 1999; Wang et al., 2012),
and predicting coastline changes (Mangor et al., 2017).
Over the past century, established works of sediment transport (e.g. Engelund & Hansen, 1967; Julien, 2010; Soulsby,
1997; van Rijn, 1993) has illustrated total sediment transport from a general perspective including both coarse and
fine components. However, recently Ma et al. (2017) reports exceptionally higher sediment transport rate in Huanghe
(also known as Yellow River; a FGDS) linked with its fine bed than that in coarse bedded flumes (Guy et al., 1966).
Starting from the Engelund-Hansen sediment transport formula (Engelund & Hansen, 1967) founded on the same
flume data set, Ma et al. (2017) derive a generalized Englund-Hansen (GEH) formula of suspended sediment transport
based on energy conservation theory, excluding the wash load, the fraction of suspended load that almost does not
communicates with local bed and flow (Chien & Wan, 1999):
$$C_D q_s^* = \alpha \theta_b^n \qquad\qquad (1)$$
where $C_D$ is the total bed drag coefficient; $q_s^* = \frac{q_s}{\rho_s \sqrt{(s-1)g d^3}}$ is the dimensionless sediment transport rate, i.e. Einstein
number; $q_s = S_* u h$ is the suspended sediment transport rate by mass, and $S_*$ is the vertically averaged suspended
sediment concentration (SSC); $\theta_b = \frac{\tau_b}{(\rho_s - \rho)g d}$ is the dimensionless total bed shear stress, i.e. Shields (1936) number;
$\tau_b = \rho C_D u^2$ is the total bed shear stress; $\alpha$ is the coefficient of $\theta_b$, and $n$ is the exponent of $\theta_b$; $s = \rho_s / \rho$ is the specific
gravity of sediment grains; $g$ is the gravitational acceleration; $c$ is the total sediment concentration by mass.
By linking Huanghe and the flume data with the GEH formula, Ma et al. (2017) find similar $\alpha$ and $n$ values for distinct
zones of $d$ (for Huanghe data, $d < 130$ μm, $\alpha = 0.895$, $n = 1.678$; for the flume data, $d > 190$ μm, $\alpha = 0.0355$, $n = 3.0$);
these zones containing data points representing similar sediment transport behaviors ($\alpha$ and $n$ values) can be identified
as sediment transport regimes. In between lies a narrow transition zone, where exceptionally high sediment load is
initiated as $d$ becomes finer. Furthermore, they suggest that such phenomenal sediment transport is associated with
the absence of dune by relating sediment transport regimes to bathymetry data of lower Huanghe ($d = 90$ μm, low
bedform height) and lower Mississippi River ($d = 280$ μm, significant dune presence).
Notwithstanding these recent advances, a quantitative theoretical interpretation of the relationship between sediment
transport regimes and prevailing bedforms is still absent, yet achievable through parameterizing the relationship
between bedform geometry and sediment transport rate. Unlike preceding semi-empirical ways, we try to interpret the
mentioned problems with sediment dynamic theories that bridge the gap between bedform prediction and sediment
transport modeling. In this paper, we first set up a sediment transport model based upon van Rijn (2007a) bedform
roughness predictor, then analyze the model calculation results to classify sediment transport behaviors into two
regimes and a transition zone, and finally discuss the causes of the regime shift in sediment transport associated with
bedform changes.





**2 Methods**
2.1 Theories
In order to estimate the sediment transport in FGDSs associated with bedform changes, a numerical model is set up
to calculate values of variables in the GEH formula, so as to explore the relationship between the suspended sediment
transport rate $q_s$, a proper approximation of total sediment flux when $d < 250$ μm (van Rijn, 2007a), and dimensionless
total bed shear stress $\theta_b$.
The suspended sediment transport rate $q_s = S_* u h$ is controlled by depth-averaged flow speed $u$ and SSC $S_*$, which is
particularly governed by their vertical profiles. Based on the basic assumptions that (1) currents are the sole driving
force of sediment transport, (2) flows over the bed are unstratified, and (3) suspended sediment transport dominates
total sediment transport when $d < 250$ μm, the logarithmic law of the wall:
$$U(z) = \frac{u_*}{\kappa} \ln\left(\frac{z}{z_0}\right) \tag{2}$$

and the Rouse (1937) profile:
$$\begin{cases} c(z) = c_a \left(\frac{z}{z_a} \frac{h - z_a}{h - z}\right)^{-b} \\ b = \frac{w_s}{\kappa u_*} \end{cases} \tag{3}$$

are utilized in this model to derive $u$ and $S_*$, directing to the final estimate of total sediment transport rate.
In the law of the wall, $U(z)$ stands for the horizontal flow speed at height $z$ to the bed, $u_* = \sqrt{\tau_b/\rho} = \sqrt{C_D} \cdot u$ is the
friction velocity, $\kappa = 0.4$ denotes the von Kármán constant, and $z_0$ refers to the total roughness length. In the Rouse
profile, $c(z)$ symbolizes the suspended sediment concentration at height $z$ to the bed, $c_a$ signifies the reference
concentration (i.e. the SSC at reference height $z_a$) by mass; $b$ represents the Rouse number, which is decided by $u_*$
and $w_s$, the settling velocity of bed sediment.
Being the average of $U(z)$, $u$ is linked to $C_D$ (related to $u_*$ and $z_0$) and $\tau_b$ (related to $u_*$). Similarly, $S_*$ is associated
with the ripple roughness height $k_{s,r}$ (related to $z_a$), $d$ (related to $c_a$ and $w_s$), $C_D$ (related to $u_*$), $\tau_b$ (related to $u_*$), and
skin bed shear stress $\tau_{bs} = \rho C_{Ds} u^2$ (related to $c_a$). Given vertically averaged flow speed $u$, median bed grain size $d$,
and water depth $h$, we still need to figure out total bed drag coefficient $C_D$, skin bed drag coefficient $C_{Ds}$, and ripple
roughness height $k_{s,r}$ to finish the calculation of sediment transport rate.
$C_D$ is a function of total bed roughness height $k_s = 30 z_0$ and water depth $h$ (Soulsby, 1997):
$$C_D = \left[\frac{\kappa}{1 + \ln\left(\frac{z_0}{h}\right)}\right]^2 = \left[\frac{\kappa}{1 + \ln\left(\frac{k_s}{30h}\right)}\right]^2 \tag{4}$$

Likewise, $C_{Ds}$ is a function of grain roughness height $k_{s,g} = 2.5 d$ (Nikuradse, 1933) and water depth $h$ (Soulsby, 1997):
$$C_{Ds} = \left[\frac{\kappa}{1 + \ln\left(\frac{k_{s,g}}{30h}\right)}\right]^2 = \left[\frac{\kappa}{1 + \ln\left(\frac{d}{12h}\right)}\right]^2 \tag{5}$$



$k_{s,g}$ and $k_s$ symbolize bed friction from different perspectives. The grain roughness $k_{s,g} = 2.5d$ is only related to the
grain size of bed sediments, referring to skin friction on the bed, whereas the total bed roughness height $k_s$ is estimated
in relation to bedform size, a function of the mobility parameter $\Psi = \frac{u^2}{(s-1)gd}$ (Manohar, 1955) and water depth $h$ (van
Rijn, 2007a).
$k_s$ is composed of three components, namely ripple roughness height $k_{s,r}$, megaripple roughness height $k_{s,mr}$, and dune
roughness height $k_{s,d}$ (van Rijn, 2007a). In this study, as the mobility parameter $\Psi$ increases, $k_{s,r}$ was linearly weakened
from $150d$ to $20d$, while both $k_{s,mr}$ and $k_{s,d}$ first grow from zero and then decrease. Subsequently, when $\Psi$ is very large
(over 600), $k_{s,mr}$ remains $0.02f_{fs}$ ($f_{fs}$ denotes fine sand factor. For $d \geq 100$ μm, $f_{fs} = 1$; for $d < 100$ μm, $f_{fs} = 10000d$), a
value usually larger than $k_{s,r}$ by an order of magnitude, whereas $k_{s,d}$ is cleared. In this regard, $k_{s,d}$ is normally
predominant in $k_s$ when $\Psi$ is small, but no longer exists when $\Psi \geq 600$.
In addition, $k_{s,g}$ and $k_s$ interact with the flow in different ways. Determined by $k_{s,g}$, the skin portion of bed shear stress
$\tau_{bs} = \rho C_{Ds} u^2$ directly initiates sediment movement and suspension. Meanwhile, with a considerable input from form
drag, $k_s$ decides the total bed drag coefficient $C_D$, which (1) significantly increases total bed shear stress $\tau_b = \rho C_D u^2 =$
$\rho u_*^2$ by directing its majority to balancing bedform drag, and (2) motivates vertical distribution of turbulence, which
resists vertical stratification and diminishes the Rouse number $b$. In this regard, changes in the mobility parameter $\Psi$
lead to different bedforms, which furthermore affect sediment transport rates.
For simplicity, our detailed algorithm is listed in Supporting Information S1 for readers' reference.

2.2 Model settings
For current-induced sediment transport in FGDSs, van Rijn (2007a) summarizes that $q_s$, the transport rate of suspended
load, is larger by one order of magnitude than $q_b$, the transport rate of bedload, as long as the grain size $d$ of bed
sediment does not exceed 250 μm (2.0 φ). To underscore the role suspended sediment transport plays in FGDSs, we
set the upper boundary of bed sediment grain size $d$ as 250 μm (2.0 φ), so that $q_s$ will remain a good approximation of
the total sediment flux. As a non-cohesive modeling approach, the lower boundary of $d$ here is placed at 62.5 μm (4.0
φ), the tipping point between sand and silt. Thus, in this numerical study, the grain size of bed sediment, $d$, ranges
from 4.0 φ to 2.0 φ, with step length 0.1 φ.
In the same time, covering scenarios in real-time fluvial and coastal settings, the water depth $h$ is continuously doubled
from 0.3125 m to 20 m, and the vertically averaged horizontal flow speed $u$ is increased from 0.5 m/s to 1.5 m/s at a
0.1 m/s step size.

**3 Results**
With the above settings, we calculate ripple roughness height $k_{s,r}$, megaripple roughness height $k_{s,mr}$, dune roughness
height $k_{s,d}$, total bed roughness height $k_s$, total bed drag coefficient $C_D$, dimensionless total bed shear stress $\theta_b$, and



dimensionless sediment transport rate $q_s^*$ for each case that combines specific median grain size $d$, water depth $h$, and
flow speed $u$. These calculation results are saved in Data Set S1.
To highlight the importance of regime shift in sediment transport, we present a log-log plot, featuring the relationship
between $C_D q_s^*$ (y-axis), the product of total bed drag coefficient $C_D$ and dimensionless sediment transport rate $q_s^*$, and
the dimensionless total bed shear stress $\theta_b$.

In Figure 1, data points on the same straight line share identical exponent $n$ and coefficient $\alpha$ of dimensionless bed
shear stress $\theta_b$ in the GEH formula ($C_D q_s^* = \alpha \theta_b^n$), thus belong to a specific sediment transport regime. Based on this
conclusion, data points are therefore categorized into dune dominant and (mega-)ripple dominant sediment transport
regimes, according to their different transport behavior (as marked in ovals in each graph) and their predominant
component of $k_s$ (see Data Set S1); typical sediment transport behavior in FGDSs corresponds with the (mega-)ripple
dominant regime (pink ovals).$\alpha$ and $n$ are subsequently calculated for both dune and (mega-)ripple dominant regimes
in each case; they are listed in Data Set S1 as well. Sandwiched by these two regimes is the narrow coexistence zone,
where sediment transport behavior is influenced by both regimes (Lapotre et al., 2017) and undergoes notable changes.

**4 Discussion**
4.1 The predominant bedform category of a sediment transport regime
As van Rijn (2007a) and our calculation suggest, we set $\Psi = 400$ and $\Psi = 600$ as criteria for defining bedform categories
(Figure 2). In the dune region ($\Psi < 400$), the dune roughness height $k_{s,d}$ is predominant in the total bed roughness
height $k_s$, whereas $k_{s,d}$ diminishes rapidly in the transition zone ($400 \leq \Psi < 600$) and ultimately stays zero in the
(mega-)ripple region ($\Psi \geq 600$) (Figure 3); the megaripple roughness height $k_{s,mr}$ takes control of $k_s$ then.
Based on the classification of bedform categories, we further propose related cut-off points for sediment transport
regimes. For typical flow speed values ($u \in [0.5, 1.5]$ (m/s)) in fluvial and coastal environments, if a particular
bedform category prevails (the contribution of such bedform data points counts for more than 50% on a grain-size-
fixed bed), then the corresponding sediment transport regime is referred to this type of bedform. Hence, $d = 3.22$ φ
and $d = 2.63$ φ are identified as tipping points of sediment transport regimes (Figure 2).
Chien et al. (1987) took Yellow River (Huanghe) as an example and notice that, in FGDSs, bedform drag can be far
greater than the skin part of total bed friction in lower flow regime, and will diminish considerably to almost zero in
upper flow regime. As suggested above, sediment transport regimes are closely associated with the predominant
bedform category. In the dune region ($\Psi < 400$, i.e. lower flow regime or coarse bed), $k_{s,d}$ upholds a considerable
weight (usually more than 50%, Figure 3) in $k_s$, leading to a larger total bed drag coefficient $C_D$ and dissipating the
majority of total bed shear stress $\tau_b$ to overcoming significant dune friction; only a small fraction of total bed shear
stress is utilized for suspended sediment transport. However, in the (mega-)ripple region ($\Psi \geq 600$, i.e. upper flow



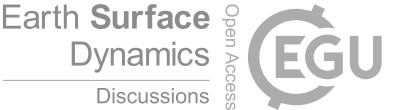

regime or fine bed, the representative setting in FGDSs), dunes are destroyed ($k_{s,d} = 0$, Figure 3) by the flow over bed,
which can reduce the total $k_s$ by up to one order of magnitude and halve the total $C_D$ (see Data Set S1). In the meantime,
the importance of grain roughness $k_{s,g}$ has increased, initiating the exceptional suspended sediment transport (Figure
1). Therefore, we suggest that increased $\Psi$ (stronger fluid flow or finer bed sediment) accelerates the degeneration of
dunes and the considerable decline in $C_D$, greatly enhancing suspended sediment transport, finally shaping the two
disparate sediment transport regimes (dune dominant and (mega-)ripple dominant).

4.2 Comparison with measured data: Importance of water depth
Derived from field survey results in the Yellow River (Huanghe, $h \approx 0.55 \sim 7.8$ m) and findings of Guy et al. (1966)'s
(GSR) flume experiments ($h \approx 0.06 \sim 0.40$ m), Ma et al. (2017) present Logistic curves, underlining sediment transport
regime shifts, i.e. changes in exponent $n$ and coefficient $\alpha$ of dimensionless bed shear stress $\theta_b$ in the GEH formula
($C_D q_s^* = \alpha \theta_b^n$), with respect to bed sediment grain size $d$. Both $n$ and $\alpha$ are indicators of bedform geometry (Engelund
& Hansen, 1967). In comparison with their results, our numerical experiments ($h = 0.625 \sim 10$ m) illustrate similar
trends in sediment transport regimes, regime shifts (coexistence zone), and estimated $n$ and $\alpha$ (Figure 4).
In our (mega-)ripple dominant regime of sediment transport ($d = 3.22 \sim 4.0$ φ), equivalent to their zone of suspended
sediment domination (Huanghe data, with $d$ finer than 2.94 φ), our calculated mean $n$ and $\alpha$ are (2.3 ~ 2.8) and (0.10
~ 0.76) respectively, while theirs are 1.678 and 0.895 correspondingly. As for our dune dominant sediment transport
regime ($d = 2.0 \sim 2.63$ φ), comparable to their sector of suspended load and bedload coexistence (GSR flume data,
with $d$ coarser than 2.40 φ), our estimated mean $n$ and $\alpha$ are (3.5 ~ 4.6) and (0.028 ~ 0.033) correspondingly, whereas
theirs are 3.0 and 0.0355 respectively. Both approaches suggest that for finer bed sediments, the exponent $n$ is smaller,
but the coefficient $\alpha$ is larger; finer beds advocate remarkable efficiency and flux of suspended sediment transport. In
view of the regime shift in sediment transport behavior, our results demonstrate a coexistence band with $d = (2.63 \sim$
3.22) φ, while they show a transition zone in $d = (2.40 \sim 2.94)$ φ.
Molinas & Wu (2001) point out the importance of water depth $h$ in the original Engelund-Hansen (EH) formula.
Derived out of Guy et al. (1966)'s flume experiment data, the original EH formula is only compliant with small water
depths ($h < 0.5$ m) and should be tested and even revised for larger $h$, due to differences in bedform development for
small and large $h$. By grouping different typical $d$, $u$, and $h$ values in FGDSs in our calculation, we compensate for the
lack of typical scenarios with different water depths in previous studies of FGDSs and furthermore demonstrate a
diverging trend in data points for increasing water depths.
Given small water depths (e.g. $h < 1$ m), dune ($k_{s,d}$) and megaripple ($k_{s,mr}$) components of total bed roughness height
$k_s$ are comparable (Figure 3 & Data Set S1), regardless of the grain size $d$ of bed sediment. Thus, although data points
within a certain prevailing bedform (dune or (mega-)ripple) can indicate similar sediment transport behavior, it is not
easy to tell apart different sediment transport regimes merely according to their data plots (Figure 1); the corresponding
regime shift as reflected by $n$ and $\alpha$ (Figure 4) is not obvious as well. But in view of rising $h$, as the dune roughness
height $k_{s,d}$ becomes prevailing in the total bed roughness height $k_s$ (Figure 3), dune dominant and (mega-)ripple





dominant sediment transport regimes commence to diverge (Figure 1, Figure 4), and the regime shift indicated by $n$
and $\alpha$ (Figure 4) is thus more apparent.
Limited by room height, the water depth of flume experiments is usually on the order of ($10^{-1} \sim 10^{0}$) m (Guy et al,
1966), whereas fluvial (Ma et al, 2017) and coastal systems (Gao & Collins, 2014) feature a typical water depth on
the order of ($10^{0} \sim 10^{2}$) m. As shown in van Rijn (2007a)'s formulae and our discussion above, the extent of bedform
development and, consequently, the suspended sediment transport behavior are strongly influenced by water depth, in
addition suggesting that a measured data set is comparable to another only if their water depths share the same order
of magnitude. Hence, it is of great necessity to take water depth $h$ into consideration in future studies of suspended
sediment transport in FGDSs by distinguishing bedforms in small and large water depths.
4.3 Future work
Our study is a preliminary numerical attempt to examine the unique sediment transport behavior of FGDSs. In reality,
due to FGDSs' high SSC, vertical stratification is amplified to a considerable extent under small $u$ (Baas et al., 2009);
even $C_\mathrm{D}$ is not vertically uniform, and the logarithmic law of the wall and Rouse profile will then no longer applicable
for the whole water column. Under this circumstance, the water column should be sliced into layers in which vertical
stratification is insignificant, and a revised (Rodi & Mansour, 1993) second-order $k$-$\varepsilon$ model can be applied to estimate
the vertical profile of flow speed (Maa et al., 2016). As $u$ increases, bolstered vertical mixing will undermine vertical
stratification, and our model can be effective in estimating total sediment transport in FGDSs.

**5 Conclusions**
With the assumptions that sediment transport is only driven by unstratified steady uniform currents and that bedload
transport is negligible, a numerical model is set up to inspect the relationship between terms in the GEH formula on
both sides of the equal sign, i.e. $C_\mathrm{D}q_s^*$ and $\theta_\mathrm{b}$. Sediment transport regimes are differentiated according to differences
in sediment transport behavior as indicated by calculation data. Between dune dominant and (mega-)ripple dominant
regimes lies the coexistence zone, the regime shift in sediment transport, which is related to the degeneration of dune
component in total bed roughness $k_\mathrm{s}$, considerably reinforcing suspended sediment transport as the flow mobility
parameter $\Psi$ increases. Additionally, greater water depth $h$ highlights such regime shift. Our study can be applied to
future modeling of sediment transport and morphological evolution.

**Code availability**
The code used in this analysis is available as a Supplement.



**Data availability**
All data used in this analysis are available as a Supplement.

**Author contribution**
QY designed the study, TZ, QY and YW performed the research, TZ and QY wrote the paper, and SG supervised
the research.

**Competing interests**
The authors declare that they have no conflict of interest.

**Acknowledgments**
The authors sincerely thank Prof. Zheng Bing Wang for his constructive comments on the original manuscript. This
study was supported by the Natural Science Foundation of China (NSFC 41676081, 41676077) and the Fundamental
Research Funds for the Central Universities (Grant No. 2016B00814).

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



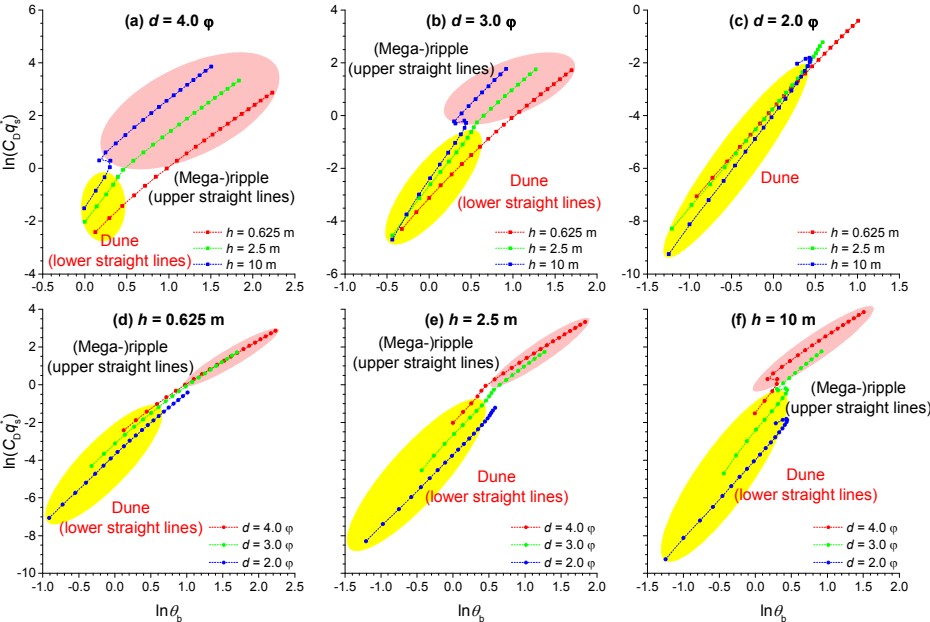



**Figure 1:** Log-log plotted relationships between y-axis: product ($C_\mathrm{D} q_\mathrm{s}^*$) of total bed drag coefficient ($C_\mathrm{D} = \left[\frac{0.4}{1+\ln\left(\frac{k_s}{30h}\right)}\right]^2$) and dimensionless sediment transport rate ($q_\mathrm{s}^* = \frac{q_\mathrm{s}}{\rho_\mathrm{s}\sqrt{(s-1)gd^3}}$, i.e. Einstein number), and x-axis: dimensionless bed shear stress ($\theta_\mathrm{b} = \frac{\tau_\mathrm{b}}{(\rho_\mathrm{s}-\rho)gd}$, i.e. Shields number), given specific combinations of typical bed sediment grain size ($d$ = 4.0, 3.0, 2.0 $\varphi$) and water depth ($h$ = 0.625, 2.5, 10 m) under fluvial, coastal, and flume settings. Data points are categorized into dune dominant (lower straight lines) and (mega-)ripple dominant (upper straight lines, associated with typical sediment transport behavior in FGDSs) regimes, and the coexistence (in-between shifts) zone, according to how the sediment transport behavior ($C_\mathrm{D} q_\mathrm{s}^*$) responds to the fluid flow ($\theta_\mathrm{b}$) through the bed. Bed sediment grain size fixed, the two regimes diverge as water depth increases (a~c). The (mega-)ripple regime tend to vanish with respect to a coarser bed, regardless of the current water depth (d~f).



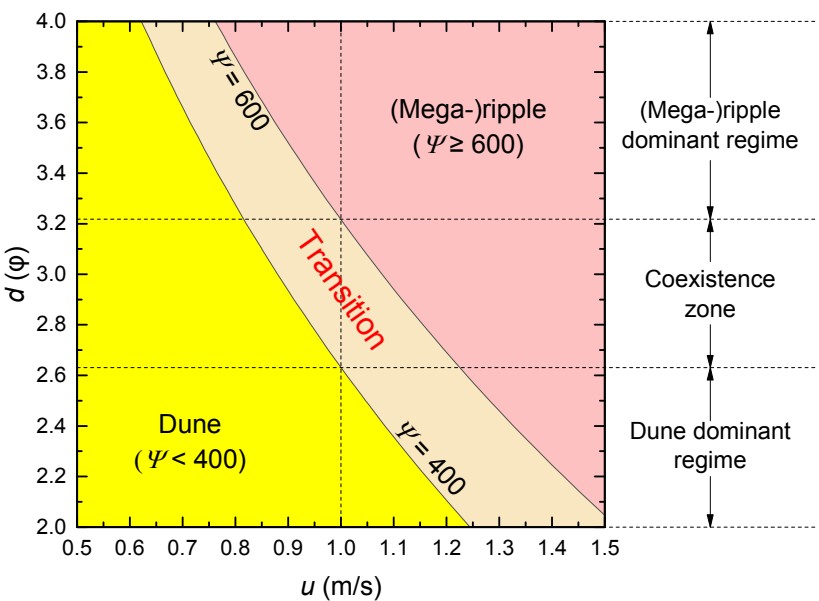

331

**Figure 2: Bedform category as a function of flow mobility parameter ($\Psi = \frac{u^2}{(s-1)gd}$). Data points with specific**

**$\Psi$ values are classified as dune ($\Psi < 400$), transition ($400 \leq \Psi < 600$), and (mega-)ripple ($\Psi \geq 600$) regions. For**

**typical vertical-averaged flow speed values ($u \in [0.5, 1.5]\ (m/s)$) in fluvial and coastal areas, sediment**

**transport over a particular grain-sized bed falls into: either a dominant regime (for (mega-)ripple dominant**

**regime, $d = 3.22 \sim 4.0\ \varphi$; for dune dominant regime, $d = 2.0 \sim 2.63\ \varphi$), as long as the contribution of**

**corresponding bedform data points exceeds 50%; or the coexistence zone ($d = 2.63 \sim 3.22\ \varphi$), when both dune**

**and (mega-)ripple points fail to become predominant (>50%).**

339




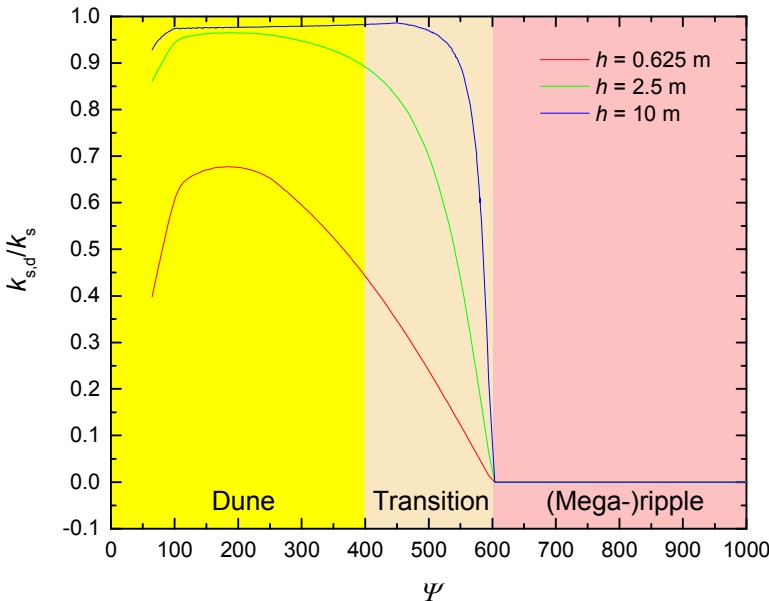

340

**Figure 3: Changes in the weight ($k_{s,d}/k_s$) of dune component ($k_{s,d}$) in total bed roughness height ($k_s$), as a function of flow mobility parameter ($\Psi = \frac{u^2}{(s-1)gd}$). Bedform categories are marked as they are in Figure 2. Under a specific water depth, the ratio $k_{s,d}/k_s$ experiences a sharp increase to reach a high stage in the dune region, then it declines hugely to 0 in the narrow transition zone, witnessing a bedform shift. As determined by the van Rijn (2007) method, this ratio remains zero in the (mega-)ripple region, indicating no dune formation above the bed. Aside from its variation with $\Psi$, the upper limit of this ratio increases rapidly as water depth goes up, exceeding 0.95 once the water depth is greater than 2.5 m.**





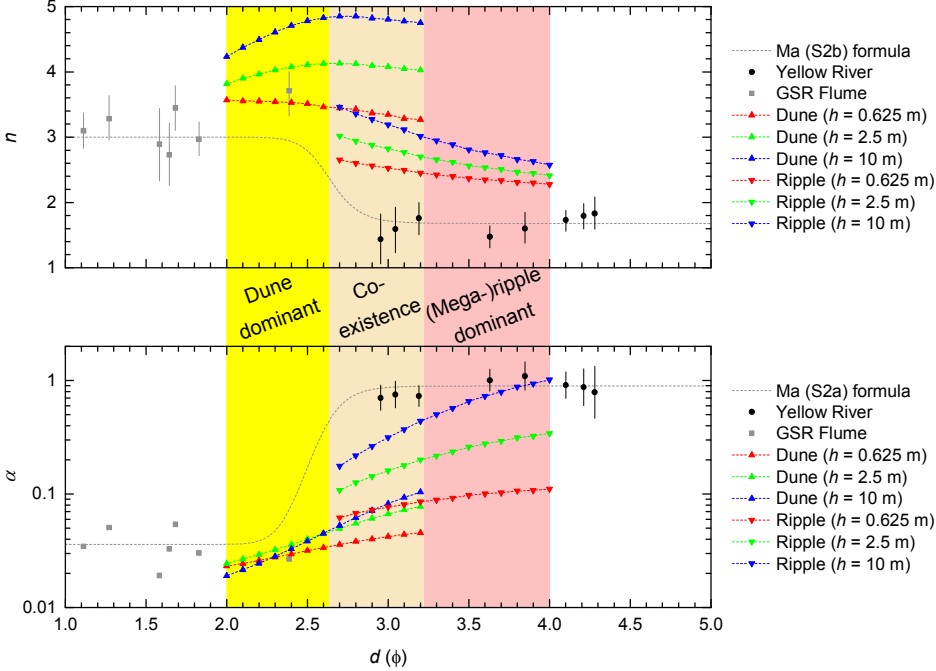

**Figure 4: Changes of y-axes: exponent ($n$) and coefficient ($\alpha$) of dimensionless bed shear stress ($\theta_b = \frac{\rho C_D u^2}{(s-1)\rho g d}$,**
**i.e. Shields number) in the Generalized Engelund-Hansen (GEH) formula ($C_D q_s^* = \alpha \theta_b^n$), with respect to x-axis:**
**bed sediment grain size ($d$). Bed regimes with respect to bed sediment grain size are marked as what they are**
**in Figure 2. As water depth ($h$) increases, data plots see increases in average $n$ (dune - 3.5 ~ 4.6, (mega-)ripple**
**- 2.3 ~ 2.8) and (mega-)ripple $\alpha$ (0.10 ~ 0.76), while average dune $\alpha$ (0.028 ~ 0.033) varies little. If relating dune**
**and (mega-)ripple points of a specific water depth, the joint curves ($n$-$d$ and $\alpha$-$d$) (1) show similar trends (almost**
**Logistic, and the regime shift/transition in the coexistence zone), as shown in Ma et al. (2017) where Logistic**
**functions are derived out of Yellow River and Guy-Simons-Richardson (1966) flume data; (2) travel upwards**
**and diverge as the water depth increases, representing a more crucial role that fluid flow ($\theta_b$) plays in shaping**
**the sediment transport behavior ($C_D q_s^*$).**