# Peer review of "Theoretical Interpretation of the Exceptional Sediment"

_Earth Surface Dynamics, 2018_

## Referee Comment (RC1) · Anonymous Referee #1 · 27 Sep 2018

The manuscript presents an analytical model to calculate the suspended load across fine-graded dispersal fluvial systems. Essentially the paper combines an existing empirical expression for the suspended transport rate (Eq. (1)) with further existing empirical expressions for the bed drag coefficient that take into account the different roughness of different sand beds (dune bed, ripple bed, moving flat bed). The keyword here is "empirical": there seems to be no first-principle-based model input from the authors, which makes me question whether the paper is sufficiently novel to justify publication in ESD. From checking the journal scope, it seems that ESD focuses on the physical processes rather than engineering-like curve fitting. I therefore believe that the manuscript would be more appropriate for an engineering journal. In favor of the authors, one

could possibly make the point that the manuscript reveals the physical mechanism behind the transition from low to large suspended load in fine-graded systems (roughness changes). However, at least for me, it has always been clear that this is the reason for this transition and I am quite sure that there have been other studies in the past making this connection (though, to be fair, I cannot point out any).

This being said, my only major criticism concerning the paper's validity is the apparent unawareness of the authors of the mechanisms that lead to the erosion of dunes when sediment becomes finer: It is quite well known that the wavelength of the smallest bedforms is controlled by the saturation length, which describes the response of the transport rate to small changes of the flow (e.g., see the review by Charru et al. 2013, doi: 10.1146/annurev-fluid-011212-140806). It is also quite well known that suspended load has a much larger saturation length than bedload (Wu et al., 2007, ISBN: 9780203938485; Claudin et al. 2011, doi: 10.1017/S0022112010005823). Now, the finer the sediment the larger the proportion of suspended load load relative to bedload and thus the larger the saturation length. This implies that bedforms with short wavelength (i.e., dunes) are eroded, which leads to the mentioned decrease of roughness. In this context, it seems inconsistent to approximate the total transport rate as the transport rate of suspended load for all fine-graded systems (assumption 3 in line 76) because the presence of dunes in fine-graded systems consisting of larger particles (but still fine) is associated with bedload transport. One could possibly argue somehow around that, but I strongly feel that something is missing here.

Minor comments: - Supplementary material should be in PDF format (I cannot read the equations without commercial software).

lines: 14 and 63: add the word "empirical" before "bed roughness predictor" 23: define phi 48: define u and h 51: "c is the total sediment concentration by mass" - the quantity c does not appear above line 51. 100: define "dune" and "megaripple"

---

## Author Comment (AC1) · 19 Oct 2018

RESPONSE: The authors appreciate the careful review by Anonymous Referee 1. Below is our response.

REFEREE: The manuscript presents an analytical model to calculate the suspended load across fine-graded dispersal fluvial systems. Essentially the paper combines an existing empirical expression for the suspended transport rate (Eq. (1)) with further existing empirical expressions for the bed drag coefficient that take into account the different roughness of different sand beds (dune bed, ripple bed, moving flat bed). The keyword here is "empirical": there seems to be no first-principle-based model input

from the authors, which makes me question whether the paper is sufficiently novel to justify publication in ESD. From checking the journal scope, it seems that ESD focuses on the physical processes rather than engineering-like curve fitting. I therefore believe that the manuscript would be more appropriate for an engineering journal.

RESPONSE: In this manuscript, the authors would like to draw our colleagues' attention to the two different sediment transport modes bridged by a regime shift associated with the abrupt bedform category change. Part of our effort in this paper is to describe the common practice of sediment transport process, from the incipient motion of a sediment grain to the behavior of the coupled grain-bed-flow system. Each step in this process is represented by one or several formulae corresponding to general physical processes, rather than simply fitted by in situ data for regional use. Hence, we believe that, though simplified and preliminary, we have been applying basic physical principles to examine a specific interaction between the Earth surface and the hydrosphere.

REFEREE: In favor of the authors, one could possibly make the point that the manuscript reveals the physical mechanism behind the transition from low to large suspended load in fine-graded systems (roughness changes). However, at least for me, it has always been clear that this is the reason for this transition and I am quite sure that there have been other studies in the past making this connection (though, to be fair, I cannot point out any).

RESPONSE: We appreciate that the anonymous referee holds a similar view on the scientific problem of sediment transport regime shift. In the manuscript, we suggest that the flow mobility parameter $\Psi$ can serve as a threshold for deciding the dominant bedform and the corresponding sediment transport mode, and higher water depth h can highlight the difference and the shift of principal bedform component and sediment transport behavior. Although these results seem empirical from the referee's view, our attempts can be the first to quantitatively examine the response of sediment transport behavior to bedform changes in the fine-grained dispersal system, and can thus assist future study in exploring the related physical mechanisms.
REFEREE: This being said, my only major criticism concerning the paper's validity is the apparent unawareness of the authors of the mechanisms that lead to the erosion of dunes when sediment becomes finer: It is quite well known that the wavelength of the smallest bedforms is controlled by the saturation length, which describes the response of the transport rate to small changes of the flow (e.g., see the review by Charru et al. 2013, doi: 10.1146/annurev-fluid-011212-140806). It is also quite well known that suspended load has a much larger saturation length than bedload (Wu et al., 2007, ISBN: 9780203938485; Claudin et al. 2011, doi: 10.1017/S0022112010005823).

RESPONSE: The authors are grateful to the referee for the opportunity to carefully examine the mechanisms of dune erosion with the suggested references above. Unfortunately, van Rijn (2007, doi:10.1061/(ASCE)0733-9429(2007)133:6(649)) did not include bedform wavelength or particle saturation length in his bed roughness predictor, which was constructed based on sediment grain size d, flow mobility parameter $\Psi$, and water depth h. Although this predictor and its initial form in roughness height (van Rijn, 1984, doi:10.1061/(ASCE)0733-9429(1984)110:10(1431)) have been widely utilized in sediment transport modeling, we believe it would be helpful to consider the relationship between particle saturation length and bed roughness in future studies of sediment transport modeling, based on our results.

REFEREE: Now, the finer the sediment the larger the proportion of suspended load load relative to bedload and thus the larger the saturation length. This implies that bedforms with short wavelength (i.e., dunes) are eroded, which leads to the mentioned decrease of roughness. In this context, it seems inconsistent to approximate the total transport rate as the transport rate of suspended load for all fine-graded systems (assumption 3 in line 76) because the presence of dunes in fine-graded systems consisting of larger particles (but still fine) is associated with bedload transport. One could possibly argue somehow around that, but I strongly feel that something is missing here.

RESPONSE: We are aware of the increasing portion of bedload in total sediment transport. However, van Rijn (2007)'s calculation suggests that the bedload transport rate is

smaller than the suspended load transport rate by at least an order of magnitude, when the bed sediment grain size d < 250 $\mu$m. Since this is from the same source where our roughness predictor comes from, we believe the transport rate of suspended load has already been a good estimate of the total sediment transport rate. It is also beneficial to include the bedload part, nevertheless, in this manuscript, the authors would like to highlight the role of suspended sediment transport in shaping dominant categories and sediment transport regimes.

REFEREE: Minor comments: - Supplementary material should be in PDF format (I cannot read the equations without commercial software). RESPONSE: The supplementary information has been converted to a PDF file.

REFEREE: lines: 14 and 63: add the word "empirical" before "bed roughness predictor" RESPONSE: Fixed.

REFEREE: 23: define phi RESPONSE: Fixed.

REFEREE: 48: define u and h RESPONSE: Fixed.

REFEREE: 51: "c is the total sediment concentration by mass" - the quantity c does not appear above line 51. RESPONSE: Fixed. We have also emphasized our usage of sediment concentration by mass in line 86.

REFEREE: 100: define "dune" and "megaripple" RESPONSE: Fixed. Please see line 103.

Please also note the supplement to this comment:
https://www.earth-surf-dynam-discuss.net/esurf-2018-64/esurf-2018-64-AC1-supplement.zip

---

## Referee Comment (RC2) · Anonymous Referee #2 · 10 Nov 2018

In this manuscript the authors propose to explain the intensity of sediment transport in what is called 'fine-grained dispersal systems' (FGDS) with a set of existing empirical relations between sediment transport, the presence of bedforms and their effect on the hydrology of the system. Although a final mechanistic explanation of this highly coupled system would be certainly welcomed, this work does not really address that because by definition, it only includes empirical correlations without a clear causal relation or predictive power (beyond the conditions for which those relations where obtained in the first place).

Even more problematic, in my opinion, is the fact that based on the title and the motiva-

tion of this study, there is no actual problem to solve. Going through the original paper by Ma et al. (2017), and the Engelund and Hansen excellent 1967 monograph in which Ma's results are based, it is clear that the central problem addressed in this paper was already solved. I mean, there is already a theoretical interpretation of the exceptional sediment transport of FGDS clearly stated in Ma et al. 2017: For fine sediments where the dominant transport mode is suspension, bedforms tend to disappear (shown by data) and the system approaches the upper-regime plane bed; in that case bedform drag is negligible and the sediment flux scales with the shear stress to power 1.5 (as in a flat bed). On the other hand, in the presence of dunes (found for pure bedload or mixed transport mode), form drag decreases the shear stress available for transport, which reduces the net sediment flux. Engelund and Hansen (1967) used rescaled experimental data to show the scaling of this effect and the implications to hydrology and sediment transport (now scaling with shear stress to a 2.5 power). Of course, their analysis contains the central effects of bedforms in an empirical form roughly equivalent to the one proposed by van Rijn (2007). Even more, in the review by Charru et al (2013) there is a potential physical mechanism for the upper-regime plane bed transition, as the characteristic wavelength of the dunes scale with the saturation length that for suspended sediment transport is very large. This essentially explains the large transport rates found in fine-sediments dominated environments.

In summary, I don't think there is enough novelty in this manuscript to justify publication, at least in the context of existing empirical formulations. Thus I recommend rejection.

In addition to those more fundamental comments, I also found the model explanation in the manuscript very difficult to follow. There are many missing equations and no clear description of the physical context where those equations fit in. Also, there is no analysis of the validity and limitations of the empirical equations; there is no proper discussion of why their model does not reproduce the empirical results of Engelund and Hansen (1967); there is no rationale for the arbitrary classification of bedforms based only on the total dimensionless shear stress (called mobility parameter) in contrast

**ESurfD**
with well-known empirical diagrams of bedform regimes; there is no definition of mega-ripples in this context (?).

**ESurfD**

Interactive
comment

---

## Editor Comment (EC1) · E. Lajeunesse (Editor) · 12 Nov 2018

Dear Authors,

We have now received two anonymous reviews of the manuscript "Theoretical Interpretation of the Exceptional Sediment Transport of Fine-grained Dispersal Systems Associated with Bedform Categories". Both reviews identify major issues which make it hard to recommend revisions to the paper. In particular, both reviewers conclude that the manuscript, based solely on empirical correlations, lacks a theoretical framework despite being in the title.

[Figure]

The topic of your manuscript is important – and the physics of bedforms development, and of their influence on the flow, has made tremendous progress over the past few years. Following the reviewers recommandation, I encourage you to reanalyse your data in the light of this work (see, for example, Charru et al. 2013, doi: 10.1146/annurev-fluid-011212-140806; Claudin et al. 2011, doi: 10.1017/S0022112010005823).

To conclude, I share the opinion of the referees that considerably more work and analysis would be required for the present manuscript to be published in E-Surf. Therefore, I recommend you not to proceed with this submission and to withdraw your paper from further processing.

Sincerely yours,

Eric Lajeunesse

---

## Author Comment (AC2) · 13 Nov 2018

Thanks for the comments from the reviewers and editor. We decided to withdraw the paper from ESD.
* * *